# Electrochemistry and Rapid Electrochromism Control of MoO_3_/V_2_O_5_ Hybrid Nanobilayers

**DOI:** 10.3390/ma12152475

**Published:** 2019-08-03

**Authors:** Chung-Chieh Chang, Po-Wei Chi, Prem Chandan, Chung-Kwei Lin

**Affiliations:** 1School of Dental Technology, College of Oral Medicine, Taipei Medical University, Taipei 11031, Taiwan; 2Institute of Physics, Academia Sinica, Nankang, Taipei 11529, Taiwan

**Keywords:** electrochromic properties, MoO_3_, V_2_O_5_, sol–gel, spinning coating

## Abstract

MoO_3_/V_2_O_5_ hybrid nanobilayers are successfully prepared by the sol–gel method with a spin- coating technique followed by heat -treatment at 350 °C in order to achieve a good crystallinity. The composition, morphology, and microstructure of the nanobilayers are characterized by a scanning electron microscope (SEM) and X-ray diffractometer (XRD) that revealed the a grain size of around 20–30 nm, and belonging to the monoclinic phase. The samples show good reversibility in the cyclic voltammetry studies and exhibit an excellent response to the visible transmittance. The electrochromic (EC) window displayed an optical transmittance changes (Δ*T*) of 22.65% and 31.4% at 550 and 700 nm, respectively, with the rapid response time of about 8.2 s for coloration and 6.3 s for bleaching. The advantages, such as large optical transmittance changes, rapid electrochromism control speed, and excellent cycle durability, demonstrated in the electrochromic cell proves the potential application of MoO_3_/V_2_O_5_ hybrid nanobilayers in electrochromic devices.

## 1. Introduction

Electrochromic properties of transition metal oxides have received much attention in recent years. The electrochromic thin films are useful for applications such as automobile, batteries, and smart windows [1,2,3,4,5,6]. Some selective inorganic electrochromic films like NiO, RuO_2_, CeO_2_, and IrO_2_ can be colored anodically, and WO_3_, MoO_3_, Nb_2_O_5_, and TiO_2_ can be colored cathodically, while some unique materials, such as V_2_O_5_, can be colored both anodically and cathodically [7,8,9]. Various hybrid materials consisting of two transition metal oxides or a transition metal oxide with organic molecules or conducting polymers, often displaying multi-electrochromism, have been developed that are suitable for various applications. Accordingly, for the fabrication of these materials, numerous methods that are economical and with minimum processing steps, such as radio frequency (RF) sputtering [10,11] and wet chemistry methods [12,13], have been adopted. Nevertheless, for the preparation of homogeneous films with a variety of precursors and on a relatively large scale, the sol–gel process stands out to be a better choice.

Among many transition metal oxides that display electrochromism, molybdenum trioxide (MoO_3_) exhibits exceptional properties [14,15,16] and therefore, modification of MoO_3_ films for the enhancement of the performance could be of interest to researchers [17,18]. The MoO_3_ film modified with a thin Ti-overlayer and V_2_O_5_-overlayer enhances the sensitivity of NH_3_ and H_2_ gas detection [18,19,20,21]. Kharade et al. prepared WO_3_/MoO_3_ hybrid by combining the sol–gel method along with thermal evaporation wherein the mixing and formation of hybrid oxide occurred during the annealing process. According to their result, although the coloration efficiency and response time of electrochromic thin films could be improved, the exact role of the MoO_3_ was still not very clear [22].

In this work, V_2_O_5_ is chosen as the constituent to form a hybrid along with MoO_3_. The V_2_O_5_ films are known to have high transparency, high electrochemical activity, and chemical stability, or in other words, the initial state of V_2_O_5_ samples have very high transmittance and large transmittance change (Δ*T*). In addition to this, the similarity between V_2_O_5_ and MoO_3_ is obvious due to their comparable ionic radii and nearly identical crystal structure. Therefore, instead of a doping or mixing process, the combination of MoO_3_-V_2_O_5_ layer structure was modified so as to capitalize their best electrochemical properties from each layer [23,24]. The same combination of oxides has been used as a catalyst for the selective oxidation of benzene and other hydrocarbons and also as cathode materials for lithium-ion batteries [25]. In the present study, the successful preparations of single-layered MoO_3_, V_2_O_5_, and MoO_3_/V_2_O_5_, V_2_O_5_/MoO_3_ hybrid nanobilayers by the sol–gel technique and their electrochromic properties have been reported. This study not only extends the scope of the potential applications of the MoO_3_/V_2_O_5_ hybrid nanobilayers but also provides the new concepts and particularly interesting applications of electrochromism.

## 2. Materials and Methods

Reagent grade Lithium perchlorate (LiClO_4_, 99%, Acros, Waltham, MA, USA), molybdenum chloride (MoCl_5_, 99%, Sigma-Aldrich, St. Louis, MO, USA), vanadium oxytrichloride (VOCl_3_, 99%, Sigma-Aldrich), isopropyl alcohol (IPA, 99.95%, Acros), hydrogen chloride acid (HCl, 99.95%, Acros), and propylene carbonate (PC, 99%, Acros) were used in this study. Figure 1 illustrates the schematic diagrams of the fabrication process of the MoO_3_/V_2_O_5_ (at an atomic ratio of 1/1) hybrid nanobilayers. Firstly, the precursors MoCl_5_ and VOCl_3_ were separately reacted with isopropyl alcohol to form their respective alkoxides, to which water was added to form sol in the presence of HCl. The sols were then spin-coated at a speed of 4000 rpm for 20 s onto the ITO pre-deposited glass substrates (15 Ω/cm^2^, RITEK Corp., Hsinchu, Taiwan) with a dimension of 2.5 × 2.5 × 0.12 cm^3^. Prior to the spin-coating process, all the substrates were rinsed in deionized water, ultrasonically cleaned in ethanol and acetone to remove organic contamination, and then dried in hot air. The as-grown MoO_3_, MoO_3_(top)/V_2_O_5_(bottom), V_2_O_5_(top) /MoO_3_(bottom), and V_2_O_5_ films were dried at 80 °C for 1 h and then heat-treated at 350 °C for 2 h under ambient atmosphere. For convenience, these aforementioned samples were named as M350, MV350, VM350, and V350, respectively. A field emission scanning electron microscope with energy dispersive spectrometer (FESEM-EDS, model S-4800, HITACHI, Tokyo, Japan) was used to examine the surface morphology, thickness, and elemental compositions of MoO_3_/V_2_O_5_ hybrid nanobilayers. The phase identification of the films was carried out by X-ray diffractometer (XRD, PANalytical X’pert PRO MPD Diffractometer, Cu Kα radiation, Almelo, The Netherlands) with a grazing incidence angle of 0.3°. The accelerating voltage and the applied current were 45 kV and 40 mA, respectively. Electrochromic properties of all samples were evaluated by cyclic voltammetry (CV, PARSTAT 2263, Perkin Elmer Instruments, Inc., Hopkinton, MA, USA), using a standard three-electrode cell system where the sample served as the working electrode, Hg/Hg_2_Cl_2_ was used as the reference electrode, and platinum as the counter electrode. The 1 M LiClO_4_ solution in propylene carbonate was used as the electrolyte. All CV measurements were performed from −1 to 0.6 V with the scan rate of 30 mV/s at room temperature. Transmittance spectra of the films were obtained by a spectrophotometer (Mini-D2T, Ocean Optics, Inc., Largo, FL, USA).

## 3. Results and Discussion

The powder XRD patterns of samples M350, MV350, VM350, and V350 are summarized in Figure 2. The diffraction peaks obtained from M350 at 12.8°, 23.4°, 25.6°, and 27.3°, resembling the (020), (110), (040), and (021) crystal planes, respectively, correspond to the orthorhombic phase of MoO_3_ (JCPDS Card No. 00-005-0508). Similarly, (020), (001), (101), and (110) crystal planes seen from the diffraction peaks of V350, appeared at 15.5°, 20.3°, 21.9°, and 26.2°, respectively, can be ascribed to the orthorhombic V_2_O_5_ (JCPDS Card No. 00-001-0359). The crystal structure of samples MV350 and VM350 belonged to the monoclinic phase (Mo_0.3_V_0.7_)_2_O_5_ (JCPDS Card No. 00-021-0576). From the above crystal studies, it is evident that the films of MoO_3_ and V_2_O_5_ hybrid have been prepared successfully through the spin-coating process followed by heat treatment. In addition to the structure, the grain sizes of the samples were calculated using Scherrer formula,
*d* = *k*λ/βcosθ(1)
where β (radians) is the full-width at half-maximum in the 2θ scan, k is a constant (0.89), λ is the X-ray wavelength (1.54 Å for Cu Kα), *d* is the particle diameter, and θ is the angle of diffraction (degrees). The calculated crystallite sizes of M350, MV350, VM350, and V350 were 27.7, 25.2, 26.5, and 34 nm, respectively.

Figure 3a,d display the surface morphologies of samples obtained from FE-SEM. Although the two hybrid nanobilayers MV350 and VM350 are heat-treated to form a new monoclinic phase (Mo_0.3_V_0.7_)_2_O_5_, it is worth noting that the morphology of MV350 is similar to M350, and that of sample VM350 is close to V350. Table 1 summarizes the semiquantitative analysis results of all samples obtained from EDS. The thickness of samples MV350 and VM350 were estimated to be approximately 100 nm from the cross-section FE-SEM images as shown in the inset of Figure 3b,c.

Electrochromism is a phenomenon in which the optical properties of a material are altered by the transferring of electrons and ions under an applied voltage. Cathodic potential leads to the migration of Li-ions along with the reduction of Mo(VI) and V(V), leading to the color change of the samples M350, MV350, VM350, and V350 into blue. Anodic potential, in contrast, leads to the oxidation and bleaching of the colored films. In the present study, it was found that the coloring process (8.2 s) was slower than the bleaching process (6.3 s), which can be explained as follows. The samples are compact and initially are in the bleached state. When a cathodic potential is applied, the compact lattice hinders the Li intercalation and results in a slow, diffusion-controlled coloring process. After intercalation, the lattice expands; therefore, a faster extraction speed is observed in the Li deintercalation process. This is the reason why the response time of the bleaching is shorter than the response time of the coloration. Figure 4 shows the second cycle of the voltammogram of samples M350, MV350, VM350, and V350. A typical CV curve of single-layered MoO_3_ and V_2_O_5_ can be easily observed in Figure 4 (black dot and green dot). Interestingly, it is apparent that the electrochemical properties of MV350 and VM350 are very similar to that of M350 and V350, respectively, which can be comprehended as the most electrochemical interactions of Li-ions occurs on the surface of the film. Additionally, since the Li-ion intercalation/deintercalation takes place within MoO_3_, the samples M350 and MV350 portray similar electrochemical behaviors. The inserted charge densities (*Q*_c_) and the calculated *Q*_c_/*Q*_a_ ratios (where *Q*_a_ is anodic charge density) for samples M350, MV350, VM350, and V350 are 23.93, 20.81, 14.82, 10.50 mQ/cm^2^ and 1, 1.01, 1.01, 1.01, respectively. It is worth noting that these *Q*_c_/*Q*_a_ values are close to one, indicating that all samples exhibit good electrochemical reversibility. During the repetitive voltammetric sweeps, apart from CV curves, even the transmittances showed good reversibility between colored and bleached states.

Figure 5a,d shows the bleaching and coloration transmittances of samples M350, MV350, VM350, and V350 between 350 and 1050 nm in 1 M LiClO_4_/PC electrolyte. It is evident that the electrochromic (EC) window of all the samples is large except for V350. This may be attributed to the coalescence of particles in the film structure during heat treatment, which led to the formation of different surface morphology and scattered or trapped the light [26]. On the other hand, although the sample V350 exhibited the poorest optical modulation as shown in Figure 5d, it is worth noting that this film with small ΔT is ideal for applications in electrochromic devices (ECDs) and smart windows [27,28]. In summary, our MV350 shows the best electrochromic Δ*T* of 22.65% and 31.4% at the wavelengths of 550 and 700 nm, respectively.

From the transmittance modulation, as seen in Figure 6, it is obvious that MV350 has the largest ΔT, indicating it is the best layer configuration for applications. From the photographs presented in the inset of Figure 6, it can be observed that the bleached state of M350, MV350, and VM350 are colorless and transparent, except for V350, which is light yellow and transparent, while, after coloration, the colored state of all the samples are near blue. Optical density (OD) and coloration efficiency (CE) are defined as follows:OD = log_10_(1/*T*)(2)
CE = ΔOD/Q_c_(3)
where *T* is the transmittance of the film at a given wavelength, ΔOD is the change in optical density upon charge insertion/extraction, and *Q*_c_ is the inserted charge density. The transmittance and electrochemical properties are summarized in Table 2. These results show that our MV350 hybrid nanobilayers possess excellent electrochromic properties and are comparable to those of V_2_O_5_ hybrid systems [29,30] prepared by pulsed spray pyrolysis techniques. These results also show that both ΔT and CE of the MoO_3_/V_2_O_5_ hybrid nanobilayers increase significantly from λ = 550 nm to λ = 700 nm and then change slightly for λ = 900 nm.

Figure 7a presents Li^+^ intercalation/deintercalation charge (Q) in the colored states and bleached states of samples M350, MV350, VM350, and V350. Figure 7b shows the intercalation during coloring time from 0 to 10 s; the amount of charge of the current density was 23.93, 20.81, 14.82, and 10.50 mC/cm^2^, while the amount of charge with deintercalation during bleaching time from 30 to 40 s was 23.83, 20.61, 14.67, and 10.42 mC/cm^2^, as shown in Figure 7c. The network-like surface morphology of VM350 obtained at annealing temperatures of 350 °C could be one of the main reasons for its low current density during both Li^+^ intercalation and deintercalation processes owing to the fact that some of the Li ions were entrapped within the network [31]. Nevertheless, the above-mentioned results determine that the MoO_3_/V_2_O_5_ hybrid nanobilayers with a rapid response time, a large optical modulation, and small charge insertion, were successfully prepared.

## 4. Conclusions

Single-layered MoO_3_, V_2_O_5_, and MoO_3_/V_2_O_5_ hybrid nanobilayers were successfully prepared by spin coating the sol–gel onto ITO/glass substrate and subsequently heat-treating at 350 °C in air. These films were tested by repetitive cyclic voltammetry that showed considerable reversibility not only in CV responses but also in optical transmittances. The single-layered MoO_3_ showed the best electrochemical property, although hybridizing with V_2_O_5_ may decrease the overall inserted charge quantity. The MoO_3_/V_2_O_5_ hybrid nanobilayers with the thickness of 100–120 nm showed the best electrochromic (EC) window with an optical transmittance changes (ΔT) of 22.65% and 31.4% at the wavelengths 550 and 700 nm, respectively. In addition, the rapid response time of the EC window was found to be about 8.2 s for coloration and 6.3 s for bleaching. This shows the potential applications of the above materials by simply modifying the layer so as to capitalize both electrochemical and electrochromic device performances at the same time.

## Figures and Tables

**Figure 1 materials-12-02475-f001:**
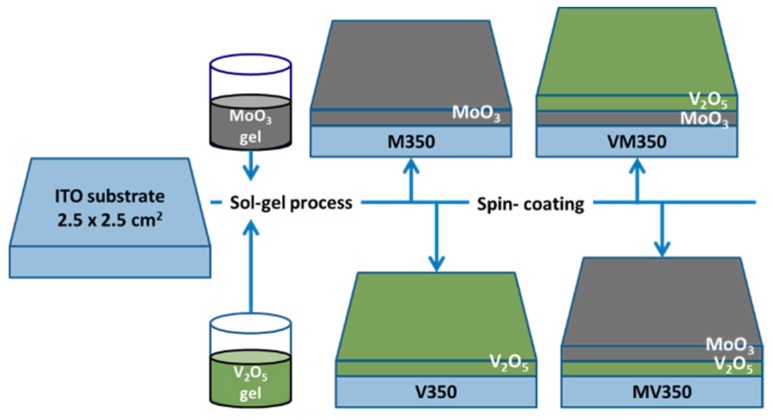
The schematic diagram of the preparation procedures for single-layered MoO_3_, V_2_O_5_, and MoO_3_/V_2_O_5_ hybrid nanobilayers.

**Figure 2 materials-12-02475-f002:**
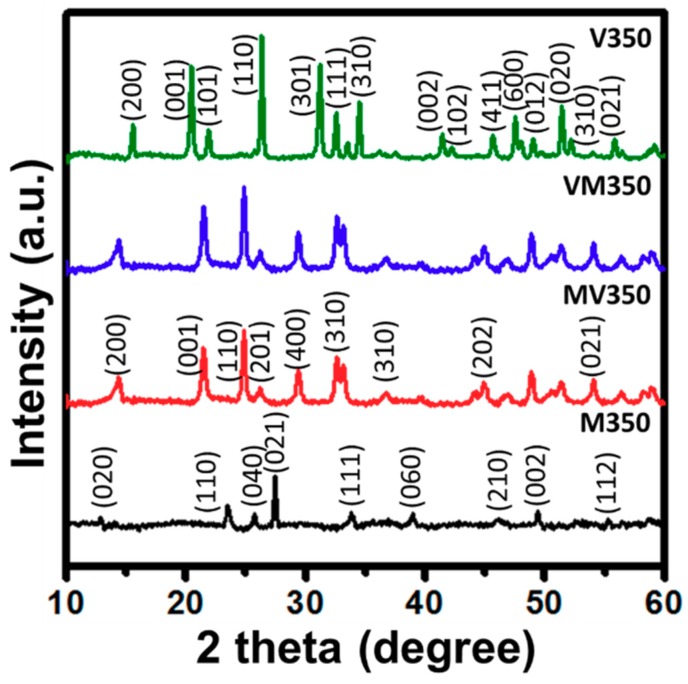
X-ray diffraction patterns of MoO_3_/V_2_O_5_ hybrid nanobilayers heated at 350 °C for 2 h in air.

**Figure 3 materials-12-02475-f003:**
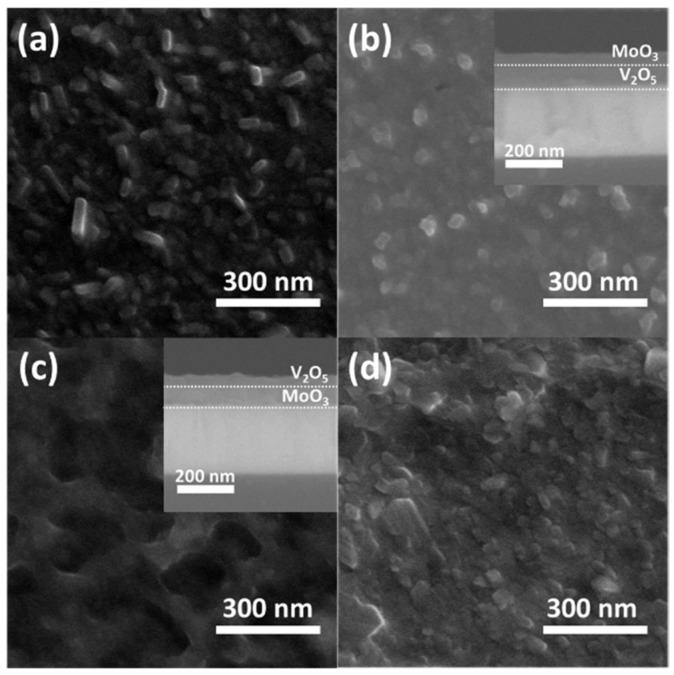
Field emission scanning electron microscope (FE-SEM) images of MoO_3_/V_2_O_5_ hybrid nanobilayers heated at 350 °C for 2 h in air. (**a**) M350, (**b**) MV350, (**c**) VM350, (**d**) V350. The inset of (**b**) and (**c**) are the cross-section images of the samples MV350 and VM350.

**Figure 4 materials-12-02475-f004:**
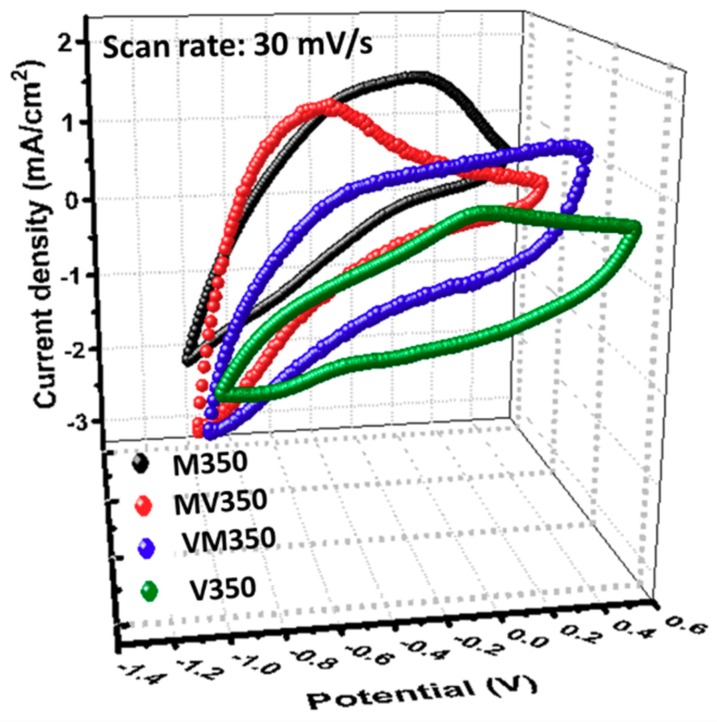
3-D cyclic voltammograms of MoO_3_/V_2_O_5_ hybrid nanobilayers heated at 350 °C for 2 h in air.

**Figure 5 materials-12-02475-f005:**
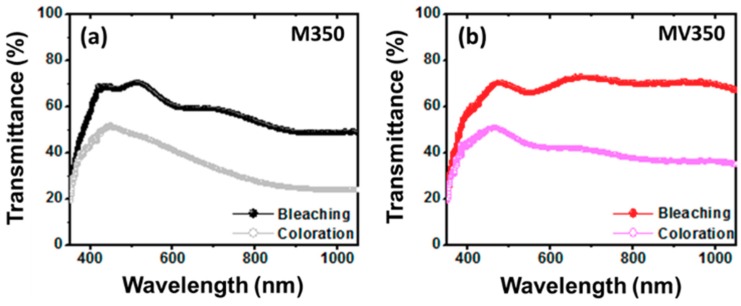
Transmittances of MoO_3_/V_2_O_5_ hybrid nanobilayers heated at 350 °C for 2 h in air. (**a**) M350, (**b**) MV350, (**c**) VM350, (**d**) V350.

**Figure 6 materials-12-02475-f006:**
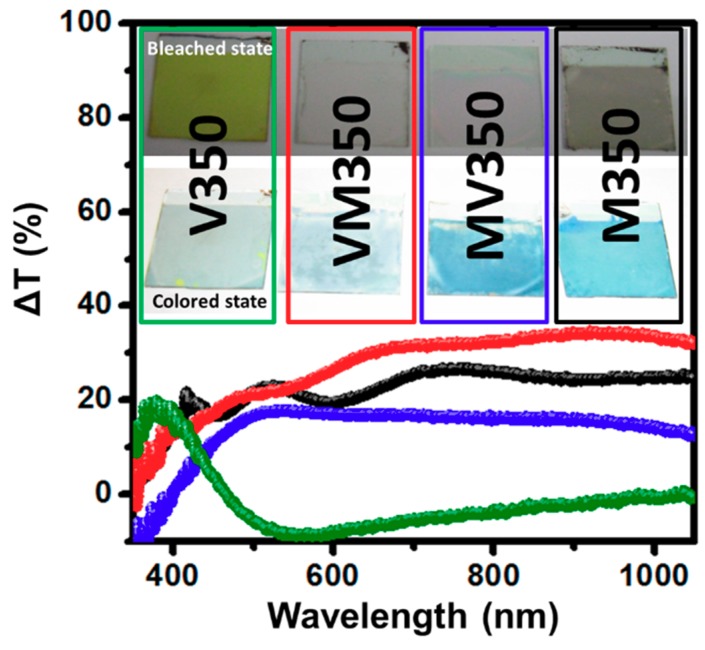
Transmittance changes (Δ*T*) of MoO_3_/V_2_O_5_ hybrid nanobilayers heated at 350 °C for 2 h in air.

**Figure 7 materials-12-02475-f007:**
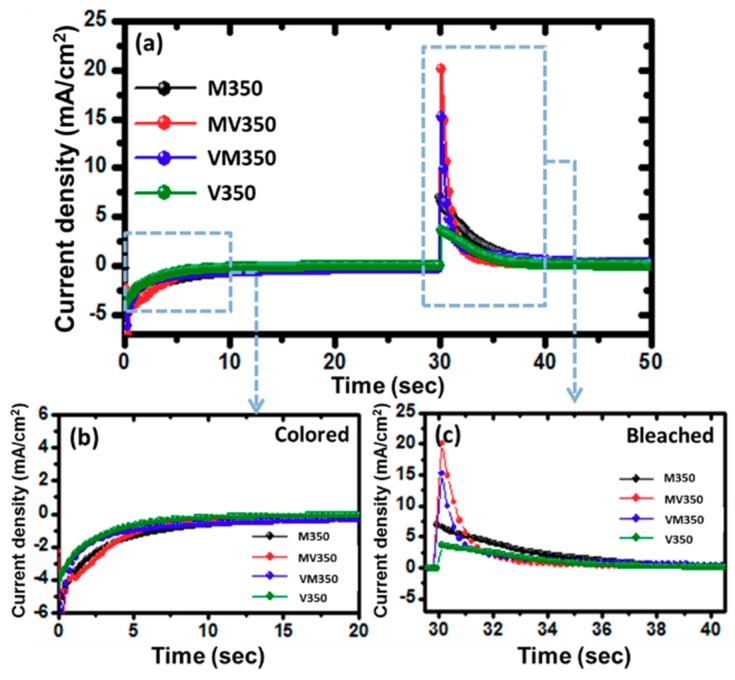
(**a**) Current density response of MoO_3_/V_2_O_5_ hybrid nanobilayers heated at 350 °C for 2 h in air. (**b**) Current density response in the colored state and (**c**) current density response in the bleached state of MoO_3_/V_2_O_5_ hybrid nanobilayers.

**Table 1 materials-12-02475-t001:** Element compositions of MoO_3_/V_2_O_5_ hybrid nanobilayers heated at 350 °C for 2 h in air.

Atomic %	Mo (L)	V (K)	In (L)	Sn (L)	O (K)	Si (K)
M350	2.43	0	18.16	1.19	67.03	11.19
MV350	0.84	2.44	15.55	0.79	71.4	8.97
VM350	1	2.55	15.29	0.93	71.22	9.01
V350	0	2.21	18.02	1	66.85	11.93

**Table 2 materials-12-02475-t002:** Electrochromic properties of MoO_3_/V_2_O_5_ hybrid nanobilayers heated at 350 °C for 2 h in air at the wavelengths 550, 700, and 900 nm, respectively. OD: optical density; CE: coloration efficiency.

Sample		M350	MV350	VM350	V350
Inserted Charge Density, *Q*_c_ (mC/cm^2^)		23.93	20.81	14.82	10.50
Extracted Charge Density, *Q*_a_ (mC/cm^2^)		23.83	20.61	14.67	10.42
*Q*_c_/*Q*_a_		1.00	1.01	1.01	1.01
λ = 550 nm	*T* _bleached_	66.72	65.96	68.36	81.36
	*T* _colored_	44.92	43.31	50.96	72.96
	Δ*T*	21.8	22.65	17.4	8.4
	ΔOD	0.1718	0.1827	0.1276	0.0473
	CE	0.0072	0.0088	0.0086	0.0045
λ = 700 nm	*T* _bleached_	58.8	72.2	70.55	80.7
	*T* _colored_	33.32	40.77	53.77	72.34
	Δ*T*	25.48	31.43	16.78	8.36
	ΔOD	0.2467	0.2482	0.118	0.0475
	CE	0.0103	0.0119	0.008	0.0045
λ = 900 nm	*T* _bleached_	48.7	70.36	69.18	80.38
	*T* _colored_	24.54	38.17	53.3	78.01
	Δ*T*	24.16	32.19	15.88	2.37
	ΔOD	0.2977	0.2656	0.1133	0.013
	CE	0.0124	0.0128	0.0076	0.0012

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
