# Peer review of "Electrochemistry and Rapid Electrochromism Control of MoO3/V2O5 Hybrid Nanobilayers"

_materials, 2019, doi:10.3390/ma12152475_

Round 1

Reviewer 1 Report

The work reports the electrochemical and electrochromic behavior of MoOand V2O5 hybrids. The reviewer considers that the article is yet not suitable for publication in Materials, requiring major revisions. In particular, the following points should be addressed:

The authors claimed that the method employed for the materials preparation was sol-gel. Nonetheless, the reviewer considers that it was not the typical sol-gel method. This aspect should be clarified.

The materials characterization by XRD revealed that the crystalline phase(s) is distinct in the bi-component layer from the individual component layers. Thus, the reviewer is not convinced that it can be claimed that the hybrid consist in (mixed ?) layers of MoO3 and V2O5 and not in one layer of a “new” mixed metal oxide. In both hybrids, the crystalline phase(s) obtained seems to be the same, and so, independent of the preparation method. It should be clarified if the peaks attribution in the hybrids (e.g. (200) and (001) diffraction planes) is coincident with the attribution in individual components diffractograms or not. Furthermore, the crystallite sizes determination should be better explained, namely indicating the diffraction peaks consider. It was assumed that the particles of metal oxides were spherical? The assumptions or limitations consider should also be pointed.

The SEM images have low resolution. The concordance between XRD and SEM results claimed in line 108 is not clear from the micrographs. Furthermore, the EDS data cannot be consider quantitative results (as claimed). Nonetheless, they could be useful to validate the existence or not of two bilayers in the hybrids, as the authors tentatively identified in Figure 3

In line 126, the explanation appointed for the differences in the coloring and bleaching processes should be validated (for example by SEM micrographs after and before the cycling process that show different porosities, or some chemical analysis that shows the lithium intercalation).

In figure 4, what is the third dimension?

In line 150, the performance of the V2O5 film obtained should be compared with the literature to validate the assumption made about their questionable application (in this particular case).

In line 152, the wavelengths selected should be justified.

By the photographs in figure 6: if the VM350 hybrid is mainly composed by V2O5 at surface (as suggested), it would not be expect that its electrochromic was more similar with the V350, namely in the oxidized state (the same for the other hybrid)? This cannot also indicate the existence of a different mixed metal oxide?

The results present in table II, namely the CE values, should be discussed, although they are very low. All the results should be compared with the literature for similar materials.

In the last paragraph, the values of switching times should be discussed (they appear in the Conclusions section but not in the discussion of the results). Furthermore, the reviewer considers that the region chosen for the evaluation of the intercalation process is not suitable; it should be used the equivalent region in a second cycle to minimize the associate error.

The reproducibility of the voltammograms during several cycles or the long-term electrochemical stability of the materials prepared was not shown, which can be critical to reclaim a possible practical application.

The scientific and English language must be polished. Namely: in page 1 (line 27), all the applications appointed require electrochromic devices, the last sentence of the first paragraph should be improved, and in line 38 “among” is repeated; in line 68, the symbol of the unit should be corrected; in line 97, the term “polycrystalline” is not correct once each oxide presents only one crystalline phase; line 197 correct “voltammetry”.

Reviewer 2 Report

The paper presented the preparation of Single-layered MoO3, V2O5 and MoO3/V2O5 hybrid nanobilayers prepared by spin coating the sol-gel onto ITO/glass substrate.   

Some questions:

 1. What is the reason of the similarity’s CV curves for V350 vs VM350, and the M350 vs MV350? Is possible that the first layer make more electrochemical interaction that the second layer and for that reason the second layer electrochemically don’t have importance? Please, explain.

2. Please include a stability-durability studies for the prepared sensors in order to check the viability of the electrochromic properties and a possible analytical use if it is possible.

3. It could be important to include a comparison table with different electrochemical characteristics in order to complete the importance and to show the best electrochemical properties of these sensors. 
4. What are the advantages of using these hybrid nanobilayers with respect other electrochromic materials? Please, include in text (in conclusions or on results and discussion sections).

5.    Eliminate the word “among” in line 38.

6.  Change the word “voltammetery” from line 197 by “voltammetry”

Reviewer 3 Report

Dear authors,

your research presented in the manuscript is very interesting and is well written. However, I recommend to improve your Discussion part - compare your results with the literature. State the novelties and differences of your work.

In addition, you can find my comments in the manuscript.

Best regards!

Round 2

Reviewer 1 Report

I don't have additional comments.